# Transfer of optical orbital angular momentum to a bound electron

Christian T. Schmiegelow[1,†], Jonas Schulz[1], Henning Kaufmann[1], Thomas Ruster[1], Ulrich G. Poschinger[1] & Ferdinand Schmidt-Kaler[1]

Photons can carry angular momentum, not only due to their spin, but also due to their spatial structure. This extra twist has been used, for example, to drive circular motion of microscopic particles in optical tweezers as well as to create vortices in quantum gases. Here we excite an atomic transition with a vortex laser beam and demonstrate the transfer of optical orbital angular momentum to the valence electron of a single trapped ion. We observe strongly modified selection rules showing that an atom can absorb two quanta of angular momentum from a single photon: one from the spin and another from the spatial structure of the beam. Furthermore, we show that parasitic ac-Stark shifts from off-resonant transitions are suppressed in the dark centre of vortex beams. These results show how light's spatial structure can determine the characteristics of light–matter interaction and pave the way for its application and observation in other systems.

[1] QUANTUM, Institut für Physik, Universität Mainz, Staudingerweg 7, 55128 Mainz, Germany. † Present address: Departamento de Física, FCEyN, UBA and IFIBA, Conicet, Pabellón 1, Ciudad Universitaria, 1428 Buenos Aires, Argentina. Correspondence and requests for materials should be addressed to C.T.S. (email: schmiegelow@df.uba.ar).

The interaction between light and matter is governed by symmetries from which conservation laws of energy, momentum and angular momentum emerge. For example, the sense of rotation of the polarization of a photon along its propagation axis determines the torque it can exert on matter. This phenomenon is known to occur on macroscopic[1,2] as well as on atomic scales[3]. At a macroscopic level, the polarization of light can be used to rotate large objects as well as micrometer-sized particles. On an atomic level, the polarization of light can change the angular momentum of bound electrons. This is expressed in the form of selection rules, which indicate which changes of angular momentum of the atom are possible when it interacts with a photon.

Specially structured light beams can have extra angular momentum associated with their spatial mode structure[4]. In particular, Laguerre–Gaussian $LG_p^l$ beams carry an additional orbital angular momentum (OAM) of $l$ per photon[5,6]. Laguerre–Gaussian beams have been proposed and employed for numerous novel applications such as high-dimensional quantum information[7], quantum cryptography[8] and quantum memories[9]. Also, OAM-carrying beams of electrons[10], neutrons[11] and electromagnetic terahertz radiation[12] have been demonstrated.

Light beams with OAM have been used to drive motion of microscopic particles in optical tweezers as well as to generate vortices in degenerate quantum gases[13,14]. Up to date, however, it was not clear if this extra angular momentum could affect the state of motion of bound electrons, that is, change the standard selection rules of optical excitation. This issue was debated during the last two decades[15–25], and first experiments[26–28] did not observe these effects.

Electromagnetically driven transitions between two atomic states occur if the superposition of their charge distributions match the multipole structure of the exciting field[29]. This allows for categorizing atomic transitions in dipolar, quadrupolar and higher orders: a dipole transition is driven by an oscillating field, a quadrupole one is driven by an oscillating field gradient and so on. Quadrupole transitions, driven by field gradients occurring in optical beams, are particulary interesting for they can even occur where there is no light intensity but only field gradient[30].

As travelling waves, optical beams have a longitudinal field gradient, which allows for driving electric quadrupole transitions[31]. A transverse gradient, due to the spatial structure of the beam front, can drive quadrupole transitions too. In particular, the centre of $LG_0^1$ beams exhibit a strong field gradient, where the intensity vanishes.

In this work, we report the observation of transfer of optical OAM from a vortex $LG_0^1$ beam to the motion of the valence electron of a trapped ion. In particular, we measure strongly modified selection rules accounting for OAM associated with the spatial structure of the beam. We observe strong excitation in the dark *penumbra* near the centre of the beam, driven solely by the transverse field gradient. In such conditions, we also observe strong suppression from parasitic ac-Stark shifts of off-resonant transitions.

## Results

### Structured beams interacting with a single trapped ion. For the experimental investigation of the interaction of structured light with the internal degrees of freedom of atoms, we use a single laser-cooled $^{40}Ca^+$ ion trapped in a microstructured segmented Paul trap with a thermal spatial spread of $\sim 60\,nm$ (see Fig. 1). Precise sub-nanometer positioning of the ion along the structured beam is achieved by changing the voltages of the trapping electrodes[32].

A continuous-wave laser near 729 nm is used to drive the $4^2S_{1/2} \leftrightarrow 3^2D_{5/2}$ quadrupole transition. As indicated in Fig. 1, this laser is shaped to the transverse $LG_0^0$, $LG_0^{\pm 1}$ and $LG_0^{\pm 2}$ modes by holographic plates[33] and focused onto the ion with a beam waist of $w_0 = 2.7(2)\,\mu m$.

To measure the interaction strength, the ion is first initialized by optical pumping on one of the $4^2S_{1/2}$ sublevels (see 'Methods' section). Upon exposure to the optical field on resonance with the quadrupole transition, the internal state of the ion undergoes coherent oscillations between the ground and excited state, which are measured by state-dependent fluorescence. The Rabi frequency $\Omega$ of these oscillations is measured to quantify the coupling strength. Each Zeeman-split sublevel of the $4^2S_{1/2} \leftrightarrow 3^2D_{5/2}$ transition is spectroscopically resolved due to an external magnetic field of 13 mT, allowing to probe all transitions $|4^2S_{1/2}, m_J = \pm\frac{1}{2}\rangle \leftrightarrow |3^2D_{5/2}, m_J = \pm\frac{1}{2}, \pm\frac{3}{2}, \pm\frac{5}{2}\rangle$ independently by tuning the laser to the respective resonance (see 'Methods' section for more details).

**Modification of transition selection rules with light OAM.** We demonstrate the joint transfer of a quantum of OAM and spin

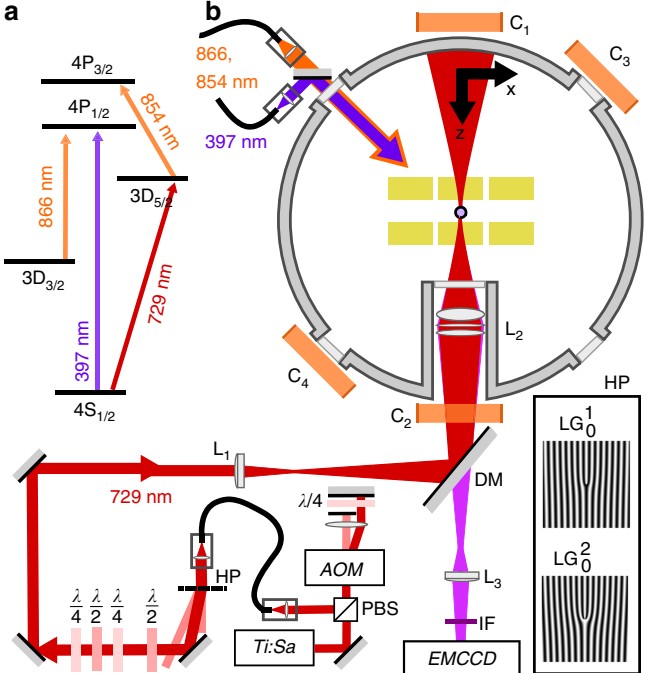

**Figure 1 | Energy levels and experimental set-up.** (**a**) Energy levels in $^{40}Ca^+$. The quadrupole transition at 729 nm is used to investigate the transfer of OAM from a photon to a single ion, the dipole transitions near 397, 866 and 854 nm are used for cooling, initialization and detection. (**b**) Experimental set-up. A single ion is trapped in a linear segmented Paul trap (yellow) inside an UHV chamber (gray). Delivered through fibres (top-left), light resonant with the dipole transitions is used for Doppler cooling, detection (397 and 866 nm) and state reset (854 nm). Resonance fluorescence near 397 nm is imaged on an EMCCD camera (bottom-right) with lenses $L_{2,3}$, passing a dichroic mirror and an interference filter. To excite the quadrupole $4^2S_{1/2} \leftrightarrow 3^2D_{5/2}$ transition, coherent light from a Ti:Sa laser is transmitted through an acousto-optic modulator for frequency and timing control, filtered by a polarization maintaining fibre and converted to the desired vortex beam with a holographic phase plate. The laser beam polarization is set by a series of quarter- and half-wave plates, and focused onto the ion by lenses $L_1$ ($f = 50\,mm$) and $L_2$ ($f = 67\,mm$). The magnetic field is controlled by coils $C_{1-4}$ plus an additional coil (not shown) in the vertical direction.

angular momentum (SAM) from the optical field to the ion. For this, we choose a setting with rotational symmetry about the propagation axis $z$ of the 729 nm beam. This is achieved by aligning the magnetic field along the $z$-direction and placing the ion in the beam centre, at the bright centre for the Gaussian $LG_0^0$ beam or at the dark *penumbra* for the $LG_0^{\pm 1}$ beams, respectively. For this geometry, the angular momentum projection along $z$ is a conserved quantity, which enforces that transitions are allowed only if the total angular momentum $m_{ph}$ of the photon matches the difference in angular momentum projection $\Delta m$ between initial and final atomic states. The photon's total angular momentum $m_{ph}$ is given by the sum of SAM (for circular polarization $\sigma_{\pm} = \pm 1$) and OAM ($l = 0$ for $LG_0^0$ and $l = \pm 1$ for $LG_0^{\pm 1}$) modes.

We verify that OAM contributes to angular momentum conservation by measuring the Rabi frequency for all possible values of $m_{ph}$ and $\Delta m$. We indeed observe coherent Rabi oscillations in all cases where angular momentum conservation is fulfilled: for $\Delta m = 0$, an OAM of $\pm 1$ compensates for a SAM of $\mp 1$, whereas for $\Delta m = \pm 2$, OAM and SAM add up. In Fig. 2, the results for initialization in the $|4^2S_{1/2}, m_J = -\frac{1}{2}\rangle$ state are shown. One clearly sees that the interaction strength for all the cases when $m_{ph} = \Delta m$ is stronger than when it is not (analogous results obtained for the opposite spin initialization are presented in the Supplementary Table 2).

For the transitions driven by the $LG_0^0$ (Gaussian) beam, the power-normalized Rabi frequencies are measured to be 13.0(8) times stronger than those driven by the $LG_0^{\pm 1}$ (vortex) beams. This is consistent with the expected relative strength[21] for the measured beam waist, as determined by the ratio of waist to optical wavelength: $\pi w_0/\lambda = 12.6(3)$. Additionally, the relative coupling strengths of different transitions are governed by the Wigner–Eckart theorem to account for coupling of SAM and OAM. For all transitions where angular momentum conservation is not fulfilled, that is, where $\Delta m \neq m_{ph}$, the measured coupling strengths are below 3% of the coupling strengths measured for the $m_{ph} = \Delta_m = \pm 1$ transitions, consistent with our error estimations (see 'Methods' section).

For the case of an $LG_0^{\pm 2}$ beam, we observe—within our experimental precision—negligible excitation for all transitions

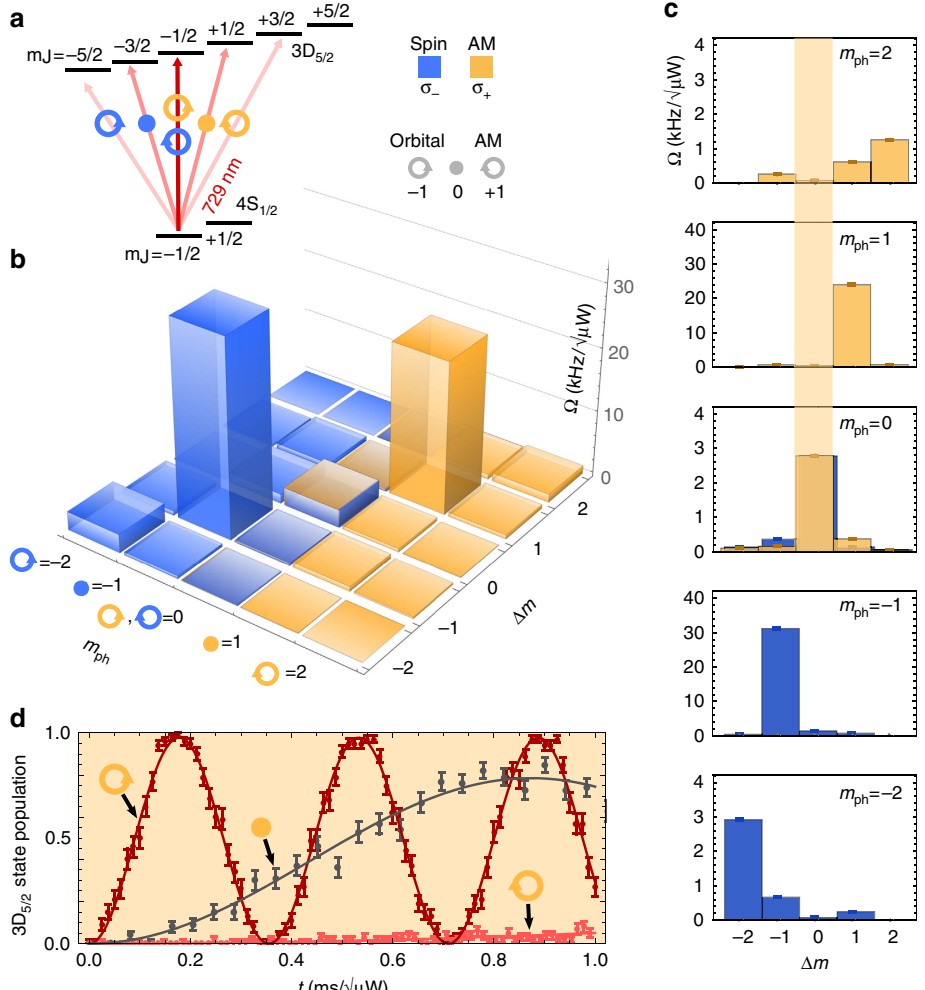

**Figure 2 | Transition selection rules.** (**a**) Energy-level structure for the $4^2S_{1/2} \leftrightarrow 3^2D_{5/2}$ manifold, indicating which field structure and polarization drives each transition. Yellow (blue) indicates polarization $\sigma_{+(-)}$; clockwise (anticlockwise) curl indicates vortex $LG_0^{+1}(LG_0^{-1})$ beams, dot indicates a Gaussian $LG_0^0$ beam. (**b,c**) Interaction strength as Rabi frequencies for all spin and OAM combinations and for all possible transitions, normalized to 1 μW of laser power. (**d**) Example measurement data corresponding to $\sigma_+$, $\Delta m = 0$ and all three beam combinations, indicated by the yellow stripe in (**c**). For all experiments, a total exposure time of 200 μs was scanned in steps of 5 μs for varying optical powers in the 3–60 μW range to determine the corresponding Rabi frequencies. Numerical values for these plots are available in the Supplementary Table 1. The corresponding raw data is available as supplementary material and its evaluation is detailed in Supplementary Note 1.

and polarization combinations. This is due to the fact that at the *penumbra* of the $LG_0^2$, the field amplitude increases quadratically in the radial direction. Thus, the quadrupole transition—which is driven by field gradients—cannot be excited at the centre of an $LG_0^{\pm 2}$ beam.

**Beam profiles**. We use the ion as a localized field probe[34,35] to map out the transverse and longitudinal field gradients for the $LG_0^0$, $LG_0^1$ and $LG_0^2$ beams by transversely moving the ion through the focus, see Fig. 3. Here we choose the magnetic field at 45° with respect to the propagation direction $z$ of the 729 nm beam, which is now linearly polarized. We probe the $\Delta m = \pm 1$ transition, where the excitation mechanism is controlled by the optical polarization[21] for this setting: if the electric field is vertically ($V$) polarized (orthogonal to the plane spanned by magnetic field and beam propagation direction), the excitation strength is determined by the longitudinal gradient, proportional to the local intensity. By contrast, for horizontal ($H$) polarization, the coupling strength is given by the transverse gradient.

For each of the LG beams, we observe that the Rabi frequency measurement with $V$ polarization reproduces the respective field amplitude profile, that is, the coupling strength is proportional to the square root of the laser intensity. Conversely, for the case of the measurements with $H$ polarization, the strength of the transverse gradient is mapped out. At the *penumbra* of the $LG_0^1$ vortex beam (Fig. 3b), the coupling mediated by the transverse gradient exceeds the one corresponding to the longitudinal gradient case by 6 s.d., that is, the ion is actually excited in the dark. We name these dark regions of an optical beam, where the effects of its gradient can be stronger than those due to the local intensity, the *penumbra*. This is not to be confused with the vortex or quantum core[36]. In fact, it is in the *penumbra* of the $LG_0^1$ where the $\Delta m = 0, \pm 2$ transitions were driven in the previous experiment.

By contrast, for the $LG_0^{\pm 2}$, we observe that both longitudinal- and transverse-gradient-driven excitations are below our sensitivity limit at the central *penumbra* of the beam, see Fig. 3c. These results show, as mentioned before, that at the centre of this beam both the electric field amplitude and its transverse gradient vanish.

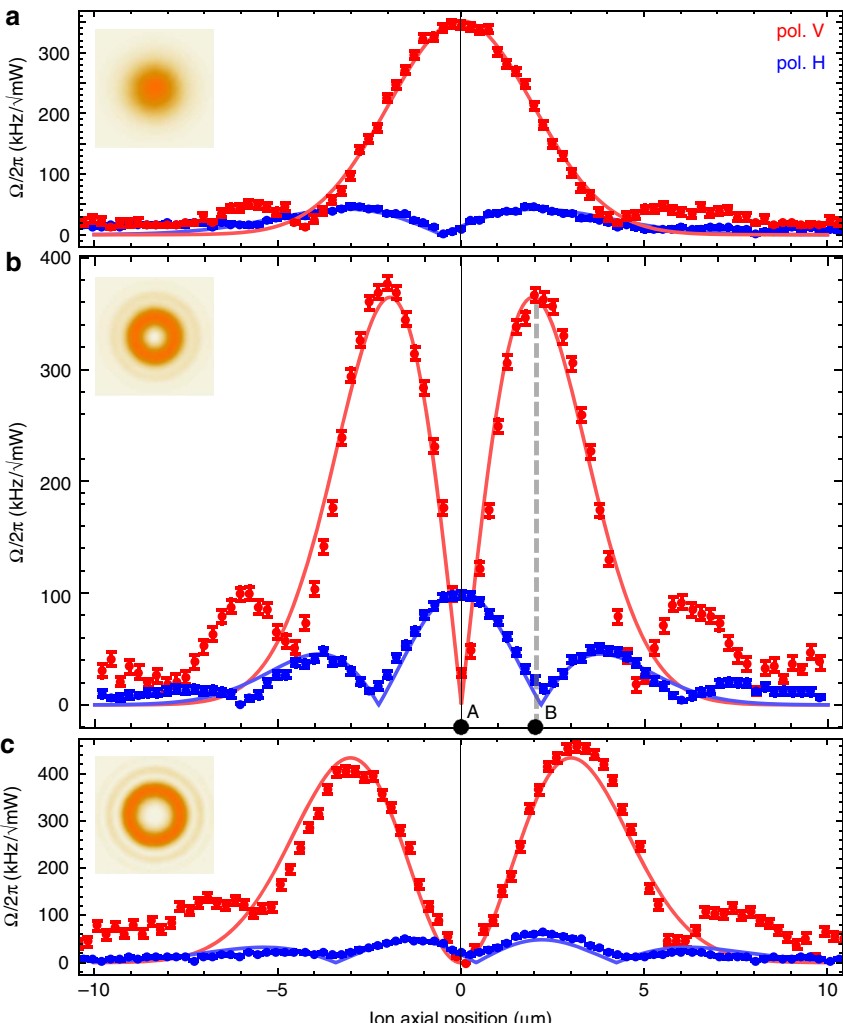

**Figure 3 | Beam profiles.** Excitation profiles in units of power-normalized Rabi frequency as a function of the ion position across three different beams: (**a**) $LG_0^0$ (Gaussian), (**b**) $LG_0^1$ and (**c**) $LG_0^2$. Red (blue) data points correspond to $V$ ($H$) polarization, that is, to excitation driven by the longitudinal (transverse) field gradient. Fits (solid lines) correspond to ideal LG beam intensity profiles (red) and their respective transverse field gradients (blue). The insets show beam intensity images taken with a CCD camera placed before the focusing lenses, revealing the same non-ideal LG outer ring structure as measured with the ion. Note that in **b**, it can be clearly seen that the ion can be excited if it is in the dark *penumbra* of the $LG_0^1$ vortex beam. The measurements of the ac-Stark shift are carried out at positions marked $A$ and $B$. Raw data as well as evaluation are can be referenced from Supplementary Note 2 and found as Supplementary Data.

The outer shoulders observed in the measurements represent deviations from ideal Laguerre–Gaussian beam profiles. We confirm this by imaging the beam profiles on a CCD camera before the focusing lenses. The corresponding outer rings are clearly observed, see insets in Fig. 3. These beam imperfections result from fibre out-coupling and diffraction on the holographic plates generating the LG beams.

For all three cases, beam-centre alignment is done with respect to the central feature of the longitudinal gradient. This feature is sharper for the $LG_0^1$ beam than for $LG_0^0$ and $LG_0^2$, leading to slight miss-alignment of the longitudinal with respect to transverse gradients for these cases.

**AC-Stark shift suppression.** An important challenge for the operation of laser-driven atomic qubits[37] as well as for the implementation of optical frequency standards[38] is the mitigation of ac-Stark shifts. These are caused by off-resonant coupling of the probe beam to parasitic atomic transitions. In our setting, the ac-Stark shifts are mainly generated either by off-resonant driving of the quadrupole transition, or by far off-resonant coupling to the $4^2S_{1/2} \leftrightarrow 4^2P_{1/2}$, $4^2S_{1/2} \leftrightarrow 4^2P_{3/2}$ and $3^2D_{5/2} \leftrightarrow 4^2P_{3/2}$ dipole transitions. The quadrupolar ac-Stark shift can be eliminated by polarization and magnetic field alignment, or by additional compensation beams. Here we demonstrate that dipolar shifts are substantially suppressed in the *penumbra* of a vortex beam, whereas the coherent coupling persists.

We measure the energy shift $\Delta_S$ caused by the 729 nm beam in the $LG_0^1$ vortex mode, which is red detuned by $\Delta = 2\pi \times 25$ MHz with respect to the $\left|4^2S_{1/2}, m_J = +\frac{1}{2}\right\rangle \leftrightarrow \left|3^2D_{5/2}, m_J = +\frac{3}{2}\right\rangle$ transition. This is done by carrying out a Ramsey experiment, where a superposition of these two states acquires a phase $\phi = \Delta_S \times t$ (ref. 37) upon exposure to the beam at power $\mathcal{P}_\Delta$ for time $t$ (see 'Methods' section). We compare the results for case $A$, where the ion is placed in the *penumbra* of the $H$ polarized beam, and for case $B$, where the ion is placed at the intensity maximum of the side lobe of the $V$ polarized beam, as marked in Fig. 3b). In case $A$, the transition is driven in the dark by the transverse gradient, whereas for case $B$, it is driven by the longitudinal gradient, and the ion is exposed to a high intensity. At position $A$, we measure an ac-Stark shift of $\Delta_S = 2\pi \times 1.54(15)$ kHz at a power of $\mathcal{P}_\Delta = 7.50(15)$ mW and a Rabi frequency $\Omega = 2\pi \times 11.93(24)$ kHz at a power of $\mathcal{P}_\Delta = 20(1)\,\mu$W. At position $B$, we measure an ac-Stark shift of $\Delta_S = 2\pi \times 19.1(1)$ kHz at a power of $\mathcal{P}_\Delta = 1.75(4)$ mW and a Rabi frequency $\Omega = 2\pi \times 15.67(31)$ kHz at a power of $\mathcal{P}_\Delta = 2.6(1)\,\mu$W.

From the measured values for the ac-Stark shift, the power-normalized quadrupolar contribution is subtracted, that is, $\Delta'_S = \Delta_S - (\mathcal{P}_\Delta/\mathcal{P}_\Omega)\Omega^2/(2\Delta)$. Additionally, the values are normalized by overall optical power $\mathcal{P}_\Delta$. The corrected, normalized shift in case $A$ is determined to be 0.70(25)% of the corresponding value for case $B$. This residual shift is attributed to thermal fluctuations of the ion position into regions of non-zero field amplitude (see 'Methods' section). Moreover, we compare the power-normalized ratios of the residual dipolar shift to the respective Rabi frequency. The quantity $\xi = (\Delta'_S \mathcal{P}_\Delta^{-1})/(\Omega\mathcal{P}_\Omega^{-1/2})$ is a suitable figure-of-merit, as it gives the ratio of the undesired shift to the desired coherent coupling when multiplied with the square root of the employed optical power. For the transverse-gradient-driven excitation in the dark, we find a suppression by $\xi_A/\xi_B = 2.5(9)\%$ as compared with the longitudinal-gradient-driven case.

## Discussion

Our results open up a realm of future research directions, in which OAM of light can be harnessed as an extra control parameter in light–matter interaction. The key aspects in our study are the use of a quadrupole transition, focusing the probe beam close to the diffraction limit and using a well-localized atomic system. Determining which of these conditions are sufficient to observe enhanced effects due to the structure of the beam is a prerequisite for extending this technique to other physical systems as well as for possible applications. These include the mapping of high-dimensional flying qubits from photons to atomic quantum memories[7], tailored interactions to improve quantum logic gates[37], the control of quantized motion of trapped particles[21,22], the suppression of ac-Stark shifts for optical clock transitions[38] and the excitation of high OAM Rydberg states[39], exciting molecule complexes[40] as well as in artificial atoms[41].

## Methods

**Optical pumping.** For the beam profile reconstructions and the ac-Stark shift experiments, where the magnetic field was at 45° with respect to the 729 nm beam, optical pumping was carried out with $\sigma_+$ polarized light driving the $4^2S_{1/2} \leftrightarrow 4^2P_{1/2}$ dipole transition near 397 nm. For the experiments on the determination of the transition selection rules, the 729 nm beam is aligned parallel to the magnetic field. Here pumping is carried out by transferring population from the $4^2S_{1/2}$ levels to be depleted to a $3^2D_{5/2}$ level, and then resetting the population to the $4^2S_{1/2}$ manifold with light resonant on the $3^2D_{5/2} \leftrightarrow 4^2P_{3/2}$ transition near 854 nm, see Fig. 1. By repeating this sequence 10 times, we can prepare the desired $\left|4^2S_{1/2}, m_J = \pm\frac{1}{2}\right\rangle$ state at a fidelity $\gtrsim 99\%$.

**Coupling strength determination sequence.** Each sequence starts with Doppler laser cooling, followed by optical pumping into either of the two ground-state sublevels $\left|4^2S_{1/2}, m_S = \pm\frac{1}{2}\right\rangle$. Next, the probe pulse near 729 nm is applied for driving Rabi oscillations between the ground state and the metastable $3^2D_{5/2}$ state. The final state is determined from observing state-dependent fluorescence on an EMCCD camera, whereas illuminating the ion near 397 and 866 nm. Observation of fluorescence indicates the ion to be in the $S_{1/2}$ state, whereas absence of fluorescence indicates a collapse into the $3^2D_{5/2}$ state. Before the sequence is repeated, we apply light near 854 nm to remove population from the $3^2D_{5/2}$ manifold, see Fig. 1. By repeating this sequence 200 times, we obtain an estimate of the $3^2D_{5/2}$ state occupation probability. By measuring this excitation probability versus the probe pulse duration, ranging up to a few hundred $\mu$s, we determine the Rabi frequency $\Omega$.

**Error estimation on the coupling strengths.** Small residual excitation measured on forbidden transitions, where $\Delta m \neq m_{ph}$, is attributed to three effects: thermal position fluctuations of the ion, imperfect optical polarization and a non-zero angle between the magnetic field and the laser propagation direction. The thermal position spread is most prominent on the transitions involving the vortex beam due to its sharp transverse structure ($m_{ph} = 0, \pm 2$, see Fig. 2c). To estimate the excess excitation, we calculate the overlap between the beam's field profile with the ion's thermal position spread. This spread is given by about 60 nm, as independently measured for our experimental conditions of a Doppler-cooled $^{40}Ca^+$ ion. Consistent with these estimations, all spurious couplings observed are below 3% of the coupling strength pertaining to the $m_{ph} = \Delta m = \pm 1$ transition.

**Stark shift determination sequence.** State preparation and read-out are carried out as for the sequence before. After preparation, the ion located either at the *penumbra* or the intensity maximum is exposed to the probe beam resonant to the $\left|4^2S_{1/2}, m_J = +\frac{1}{2}\right\rangle \leftrightarrow \left|3^2D_{5/2}, m_J = +\frac{3}{2}\right\rangle$ transition at a pulse area of $\pi/2$, such that a balanced superposition of both states is created. Next, it is exposed to the off-resonant vortex beam for time $t$, which induces an ac-Stark phase shift on the superposition. Finally, a second resonant $\pi/2$ pulse is applied. After recording the final population in the excited state versus $t$, the ac-Stark shift is determined by the frequency of the resulting coherent oscillations.

**Data availability.** The authors declare that the main data supporting the findings of this study are available within the article and its Supplementary Information files. Extra data are available from the corresponding author upon request.

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

## Acknowledgements

We thank Rupert Ursin and Anton Zeilinger for lending us the holographic phase plates; A. Wiens and A. Walther for contributions on early stages of the experiment; S. Franke-Arnold and D. Budker for useful comments; and A. Z. Khoury for inspiring this work with a lecture on orbital angular momentum of photons at the J. A. Swieca School in 2008 in Sao Paulo. C.T.S. acknowledges the support of the Alexander von Humboldt Foundation.

## Author contributions

C.T.S. and F.S.-K. conceived the idea of the experiment. Experimental data were taken by C.T.S. and J.S, using an apparatus primarily set up by C.T.S., J.S., H.K. and T.R. Data analysis was performed by C.T.S., J.S. and U.G.P. The paper was written by C.T.S., F.S.-K., J.S. and U.G.P., with input from all authors.

## Additional information

**Competing financial interests:** The authors declare no competing financial interests.

