## [Peer Review File · Nature Communications]

Reviewer #1 (Remarks to the Author):

This is a nice and very interesting work on the transfer of the orbital angular momentum of light to internal degrees of a trapped ion. In my opinion the experiment is non-trivial and the conclusions are supported by the presented data. I therefore think that this work should be published in Nature Communications. I do, however, have a few points and questions, which I hope, can be answered by the authors.

1. As a first point, the authors correctly point out the necessity of field gradients to drive quadrupole transitions and much of their work is concerned with the presence of a transverse field gradient in a vortex beam of order $l=1$, where l refers to the quantised orbital angular momentum.

Maybe it would be helpful to clarify the importance of both the field gradient and the orbital angular momentum of the light for their experiment, if the case of a Laguerre-Gaussian beam (or any vortex beam for that matter) is contrasted with a Hermite-Gaussian beam, which also has a field gradient along a preferred direction, but does not carry orbital angular momentum. In my understanding this should drive the same transitions as the $l=0$ case, albeit now with both a longitudinal and transverse gradient.

2. As a second point, I am wondering about the effect of focussing on the vectorial field distribution. The authors record the intensity pattern before focussing in the insets in Fig. 3, but the data points in the main figures trace out the intensity pattern of one polarisation component and its gradient. I am wondering whether it would not be more informative to compare the measured Rabi frequencies not with an ideal intensity profile but rather with the recorded intensity profile shown in the insets. Of course, there will be deviations, because of the focussing, which gives rise to longitudinal and potentially even cross-polarised components, but a direct comparison could yield additional information.

Moreover, in Fig 3c the ideal profile and its gradient seem to be offset with respect to each other as the maxima of the intensity do not seem to coincide with zeros of the gradient. That seem odd to me as in Fig. 3b there is near perfect match.

3. A third point which I find slightly confusing is connected to the choice of the word 'penumbra' for the dark centre of the vortex beam. From the use of this term it has not been fully clear to me if this refers to the actual singularity or an extended region around it. In the literature there have been various attempts to define a radius of the vortex core, the inner region in which atomic transition behave differently [Berry, M. V. & Dennis, M. R. Quantum cores of optical phase singularities. *J. Opt. A: Pure Appl. Opt.* 6, S178-S180 (2004); Barnett, S. M. On the quantum core of an optical vortex. *Journal of Modern Optics* 55, 2279-2292 (2008); 1. Barnett, S. M. & Berry, M. V. Superweak momentum transfer near optical vortices. *J. Opt.* 15, 125701 (2013)]. The question I have is now, whether the ion has to be positioned within a radius of the point-like singularity of the vortex in order to drive the transitions involving an exchange of orbital angular momentum. Or in other words, how far extends the penumbra of the light beam?

4. As a last point of personal preference, I do not think the title captures the essence of the interesting results presented in the manuscript. Of course I understand that there are difference of style in choosing a title, but perhaps mentioning the transfer of orbital angular momentum to internal degrees would highlight the importance of this work better.

Reviewer #2 (Remarks to the Author):

The authors use study the influence of light's orbital angular momentum (OAM), in addition to the

spin angular momentum (SAM), on the light-ion interaction strength. They found modified optical selection rules where the orbital angular momentum of photons ($L \cdot \hbar$, where the wavefronts are L times per wavelength twisted around the beam axis) takes part, simply in the $J=L+S$ picture. In my opinion, the authors produced exceptionally beautiful data in this seminal experiment; after more than 10 years of unsuccessful experiments and contradicting theoretical studies, these clearly positive results will very likely mark the start of a whole new field within quantum optics.

In more detail: The authors use a unique setup where a 40Ca^+ ion can be positioned with an accuracy better than 100 nanometers, and controlled with state of the art ion spectroscopy methods. They use resonant light with controlled spin- and orbital angular momentum to drive Rabi oscillations; the measured Rabi frequency is proportional to the coupling strength. This is done on Zeeman-split quadrupole transitions that are driven by field gradients, in contrast to intensity for dipole transitions. The authors explore and explain this issue nicely: First, they demonstrate the influence of the gradient strength, for the longitudinal gradient of fundamental-mode beams, and for transverse gradients, which appear in OAM beams. Second, they make the main discovery that, for suitable experimental configuration, the OAM of the photons determines which Zeeman transition can be addressed: Not only the gradient, but also the sign of the local OAM value is important. The five transitions have Δ_m ranging from $-2..2$. They measure that the $\Delta_m = \pm 2$ transitions are 2-3 times more likely to be excited by having $\text{OAM} = \text{SAM} = \pm 1$ than with the "wrong" optical AM. This ratio is even higher if one compares the Rabi frequencies of the $\Delta_m = \pm 1$ transitions excited with $m_{\text{ph}} = \pm 2$ to $m_{\text{ph}} = 0$ (i.e., comparing only doughnut modes). I find this a very clear result.

The authors also study how "excitation in the dark" reduces the ac-Stark shift. The discussion is very brief, but understandable. The result is that ions in the "penumbra" experience little Stark shift, which is important for metrology applications.

Discussions of the (partially contradicting) previously published theories are not included; this I find the best solution for the present case: The experimental evidence is perfectly clear, and the explanation does not require any complex theories. The present work has already stimulated revision of some theories.

The manuscript is of very high quality and very nice to read, I have only minor comments:

- * I wonder if any nonparaxial effects are important. The numerical aperture of the 729nm beam is small ($\text{NA} = 0.09$); it is correct to assume that the used experimental configuration ($B \ll \lambda$ beam axis) in combination with a quadrupole transition removes any spurious effects like spin-orbit coupling?
- * A table with the numerical Rabi frequencies for all transitions in Fig. 2c would be nice; possibly also for testing theories (might be published via some other means).
- * Is "penumbra" adequate for the vortex core? I find it a good term for the region $r > 0$ (not fully dark), but the OAM vortex ($r = 0$) is perfectly dark in paraxial optics.

Reviewer #3 (Remarks to the Author):

(A) The manuscript by Schmiegelow and co-workers reports on the experimental study of the excitation of a single trapped ion by a twisted (Laguerre-Gaussian) light. The experiment has dealt especially with the transitions between magnetic substates of the ground $4s_{1/2}$ and the excited $3d_{5/2}$ states of the singly-ionized calcium, Ca^+ . Three important results are presented in the article. (i) If the atom is placed in the center of the beam (on the vortex line), one can use the total angular momentum (TAM) of incident photons to "control" the (orbital momentum) selection rules and to induce transitions between particular magnetic substates. (ii) By moving the atom within the wave-front and observing the photo-excitation, one can visualize the intensity profile of the twisted beam. (iii) The twisted beams with central zero-intensity spot can be used to suppress

ac-Stark shifts.

(B) The present study is novel and provides an important step towards the use of twisted light beams in atomic physics. As was mentioned by authors in their abstract, the question of whether twisted radiation can "talk" to individual atoms and their electronic degrees of freedom was up to now debatable. Even though the possibility of controlling the atomic selection rules with the help of TAM was predicted theoretically (in addition to references given by authors, I would propose two very "fresh" papers: J. Phys. B: At. Mol. Opt. Phys. 49 (2016) 074007, and Phys. Scr. 91 (2016) 064001), the experimental observation of the effect was up to the present missing.

(C)-(D) Being theoretician I can not fully judge the experimental metrology and error estimates. As I can see, the present experimental study is based on the well-established Paul-trap techniques which allow storing and operating of single atoms with a high precision.

(E) The conclusions of the present study are robust and reliable. The reported experimental data (and their interpretation) well agree with the theoretical expectations. I would like to stress, moreover, that the results regarding the possibility of controlling the atomic selection rules and regarding suppressing the ac-Stark shift are of significant importance for atomic physics community. They open new ways (here I fully agree with author's conclusions) in the fields of quantum information and quantum metrology.

(F) For my opinion, the experimental data is complete and conclusive.

(G) As I mentioned already above, I would recommend authors to consider two new theoretical papers regarding atomic excitation by twisted light: J. Phys. B: At. Mol. Opt. Phys. 49 (2016) 074007, and Phys. Scr. 91 (2016) 064001. Moreover, one more interesting work is available in archive: <http://arxiv.org/abs/1602.06640> . Finally, I would like to point that in Ref. [27] some information (page number?) is missing.

(H) I would like to recommend authors some text corrections, to make manuscript more "readable" for a wider audience. In the "Introduction", I would find it useful to briefly remind the reader the basic properties of twisted beams. It would be nice to recall the helical structure of the phase front, the inhomogeneous intensity profile - to stress that "internal structure" of twisted beams is more complicated comparing to standard plane-wave light. And to stress that characteristic "size" of twisted beams (size of intensity rings, for example) is much larger than atomic size. Actually, this different-scale-argument was one of arguments in the past why doubts have arisen whether (small) atom can "talk" to (large) beam.

Another comment: could authors clearly state in the main part of the paper, not in supplementary results, about how accurate the atom can be located within the LG wave front. It applies both for Fig. 2 and Fig. 3 results. Is that only 60 nm, mentioned in "Error estimation"?

Reply to reviewers.

We thank all reviewers for their comments. They have been very constructive in making this new revision of the paper.

Reviewer #1 (Remarks to the Author):

“This is a nice and very interesting work on the transfer of the orbital angular momentum of light to internal degrees of a trapped ion. In my opinion the experiment is non-trivial and the conclusions are supported by the presented data.”

“I therefore think that this work should be published in Nature”

We thank the referee for the very positive remark and estimation of our work.

“Maybe it would be helpful to clarify the importance of both the field gradient and the orbital angular momentum of the light for their experiment, if the case of a Laguerre-Gaussian beam (or any vortex beam for that matter) is contrasted with a Hermite-Gaussian beam, which also has a field gradient along a preferred direction, but does not carry orbital angular momentum. In my understanding this should drive the same transitions as the $l=0$ case, albeit now with both a longitudinal and transverse gradient.”

This is an interesting point the reviewer raises. Hermite-Gaussian beams have a transverse field gradient but no defined orbital angular momentum. Each photon is, in fact, in a superposition of the states with + and – orbital angular momentum. In other words, a superposition of Laguerre-Gauss beams with opposite chiralities. This is analogous to the case of linear and circular polarization. If such a beam should be used, in the rotational symmetric configuration, one would expect, as correctly stated, the $l=0$ transition, but also the $l=2$ or $l=-2$ transition depending on the polarization. Moreover, if one had linear polarization and a Hermite-Gauss mode (“linear-OAM”), then all transitions $l=2,0,-2$ would be allowed.

We did not include this discussion for the following reasons: On the one hand, because we did not perform measurements with Hermite-Gaussian beams and believe it would complicate the understanding rather than clarify it. Also, if we would have included other beam types in the discussion, not only Hermite-Gaussian, but radially and azimuthally polarized beams as well as other beam combinations should have been included. This is beyond the scope of the experiments reported and will be subject of future work.

“As a second point, I am wondering about the effect of focussing on the vectorial field distribution. The authors record the intensity pattern before focussing in the insets in Fig. 3, but the data points in the main figures trace out the intensity pattern of one polarisation component and its gradient. I am wondering whether it would not be more informative to compare the measured Rabi frequencies not with an ideal intensity profile but rather with the recorded intensity profile shown in the insets. Of course, there will be deviations, because of the focussing, which gives rise to longitudinal and potentially even cross-polarised components, but a direct comparison could yield additional information.”

After initial attempts to match the measured Rabi frequency profiles to measured intensity distributions, we had decided to present the data along with ideal beam profiles for the following reasons: By means of a CCD camera measurement, we only have access to the *intensity* distribution *before* the focusing optics. The actual Rabi frequency profile at the ion location depends on the *field* distribution before the focusing, including the phase profile to which we

have no access. Furthermore, it depends on non-ideal features of the focusing optics which is also not fully characterized. Thus, inferring a Rabi frequency profile from the measured intensity profile would require adding information “by hand”, which would again complicate the discussion without providing additional insight.

Moreover we would like to stress, that the important new physics we describe in our work, occurs at the center of the beam, for which results, theoretical predictions and the previously described procedure all match.

“Moreover, in Fig 3c the ideal profile and its gradient seem to be offset with respect to each other as the maxima of the intensity do not seem to coincide with zeros of the gradient. That seem odd to me as in Fig. 3b there is near perfect match.”

The beam alignment was redone between the measurements for the two different polarizations. In all cases, alignment was first done on the V polarization, i.e. for the signal proportional to the square root intensity of the beam (red in Fig. 3). For the cases of OAM=0,2 however, the central feature is less sharp, such that the alignment to the center is less precise as for the OAM=1 case. This resulted in a slight misalignment of the center, which varied from measurement to measurement. We added a corresponding statement in the main text:

“For all three cases, beam-center alignment is done with respect to the central feature of the longitudinal gradient. This feature is sharper for the LG $_0^1$ beam than for LG $_0^0$ and LG $_0^2$ leading to slight miss-aliment of the longitudinal with respect to transverse gradients for these cases.”

“A third point which I find slightly confusing is connected to the choice of the word 'penumbra' for the dark centre of the vortex beam. From the use of this term it has not been fully clear to me if this refers to the actual singularity or an extended region around it. In the literature there have been various attempts to define a radius of the vortex core, the inner region in which atomic transition behave differently. The question I have is now, whether the ion has to be positioned within a radius of the point-like singularity of the vortex in order to drive the transitions involving an exchange of orbital angular momentum. Or in other words, how far extends the penumbra of the light beam?”

We thank the reviewer for bringing up these citations and the notion of “quantum core of an optical singularity”. It would be certainly interesting to observe such effects. However, our experiment operates far away from the proposed regime. The transition we probe has a linewidth of approximately 1 Hz, and our beam has a waist of 2-3 μ m. Using formula (8) from the Berry & Dennis paper, we calculate the quantum core of the beam to be about 0.2pm, which is even smaller than the atomic radius! Moreover, the thermal spread of the ion position in the experiments we conducted is estimated to be about 60nm.

This fact, and the results on figure 3 clearly show that the ions do not have to be positioned within a radius of the point-like singularity of the vortex to observe OAM effects on the driven transitions.

We decided to use the word “penumbra”, which indicates “a space of partial illumination (as in an eclipse) between the perfect shadow on all sides and the full light” (Merriam Webster – Dictionary) to indicate that the ion is to be considered as an object with spatial extension, such that it is still exposed to residual illumination even if placed at the optical singularity. We have extended our explanation the use of this word now in the main text in order to clarify its meaning:

“ We call the beam’s penumbra where the dark regions of an optical beam where the effects of its gradient can be stronger than those due to the local intensity”

“As a last point of personal preference, I do not think the title captures the essence of the interesting results presented in the manuscript. Of course I understand that there are difference of style in choosing a title, but perhaps mentioning the transfer of orbital angular momentum to internal degrees would highlight the importune of this work better.”

We thank the reviewer for the suggestion. We propose the title “Transfer of optical orbital angular momentum to a bound electron”.

Reviewer #2 (Remarks to the Author):

“The authors use study the influence of light's orbital angular momentum (OAM), in addition to the spin angular momentum (SAM), on the light-ion interaction strength. They found modified optical selection rules where the orbital angular momentum of photons ($L\hbar$, where the wavefronts are L times per wavelength twisted around the beam axis) takes part, simply in the $J=L+S$ picture. In my opinion, the authors produced exceptionally beautiful data in this seminal experiment; after more than 10 years of unsuccessful experiments and contradicting theoretical studies, these clearly positive results will very likely mark the start of a whole new field within quantum optics.”

“The manuscript is of very high quality and very nice to read”

We thank the referee for the positive remarks.

“I wonder if any nonparaxial effects are important. The numerical aperture of the 729nm beam is small ($NA=0.09$); it is correct to assume that the used experimental configuration ($B \ll$ beam axis) in combination with a quadrupole transition removes any spurious effects like spin-orbit coupling?”

In the present work, no effects of strong focusing were observed. We have chosen the beam size to be in a regime where transverse-gradient effects could be observed, but the focusing is clearly not strong enough to observe spin-orbit coupling [EJPD 66(6), 2012]. Future research will be conducted into this very interesting direction using strongly focused beams.

* A table with the numerical Rabi frequencies for all transitions in Fig. 2c would be nice; possibly also for testing theories (might be published via some other means).

The table with the Rabi frequencies as well as the data corresponding to figure 2 is now provided as supplementary material. Extra supplementary material is provided for data with the $m=+1/2$ initialization that is not plotted. The supplementary material contains also raw data of Fig. 3.

* Is "penumbra" adequate for the vortex core? I find it a good term for the region $r>0$ (not fully dark), but the OAM vortex ($r=0$) is perfectly dark in paraxial optics.

We do not use “penumbra” as a synonym for the vortex core. We refer the Reviewer to the discussion with Reviewer #1 on this issue. We have extended our explanation the use of this word now in the main text in order to clarify its meaning.

“ We call the beam’s penumbra where the dark regions of an optical beam where the effects of its gradient can be stronger than those due to the local intensity”

Reviewer #3 (Remarks to the Author):

“The manuscript by Schmiegelow and co-workers reports on the experimental study of the excitation of a single trapped ion by a twisted (Laguerre-Gaussian) light. The experiment has dealt especially with the transitions between magnetic substates of the ground $4s_{\{1/2\}}$ and the excited $3d_{\{5/2\}}$ states of the singly-ionized calcium, Ca^+ . The present study is novel and provides an important step towards the use of twisted light beams in atomic physics.”

We thank the third referee for the positive remarks.

“I would recommend authors to consider two new theoretical papers regarding atomic excitation by twisted light: J. Phys. B: At. Mol. Opt. Phys. 49 (2016) 074007, and Phys. Scr. 91 (2016) 064001. Moreover, one more interesting work is available in archive: <http://arxiv.org/abs/1602.06640> . Finally, I would like to point that in Ref. [27] some information (page number?) is missing.”
We thank the Reviewer for the new references. J. Phys. B: At. Mol. Opt. Phys. 49 (2016) 074007 has been now included in the Discussion. Phys. Scr. 91 (2016) 064001, and <http://arxiv.org/abs/1602.06640> included in the list of theory papers discussing OAM-Atom theory Reference 27 has been corrected.

“I would like to recommend authors some text corrections, to make manuscript more "readable" for a wider audience. In the "Introduction", I would find it useful to briefly remind the reader the basic properties of twisted beams. It would be nice to recall the helical structure of the phase front, the inhomogeneous intensity profile - to stress that "internal structure" of twisted beams is more complicated comparing to standard plane-wave light. And to stress that characteristic "size" of twisted beams (size of intensity rings, for example) is much larger than atomic size. Actually, this different-scale-argument was one of arguments in the past why doubts have arisen whether (small) atom can "talk" to (large) beam.”

The introduction was substantially re-written taking into account the Reviewer's observations and the nature communication format requirements.

“Another comment: could authors clearly state in the main part of the paper, not in supplementary results, about how accurate the atom can be located within the LG wave front. It applies both for Fig. 2 and Fig. 3 results. Is that only 60 nm, mentioned in "Error estimation"?”

The extent of the ion's thermal position distribution is now clearly stated in the text.

Reviewer #1 (Remarks to the Author):

I would like to thank the authors for their comprehensive reply to the comments of all referees.

I am still slightly puzzled how the ion experiences a transition involving orbital angular momentum if does not have to be placed within a region that contains the singularity, but perhaps this issue will indeed become clearer in subsequent work as suggested by the authors.

Apart from this point I think the authors have addresses all the comments from the referees and I can recommend publication in Nature Communications.

Reviewer #2 (Remarks to the Author):

In my opinion, the authors have answered all questions raised in the previous round satisfactorily. As I wrote before, I find this is a very nice piece of work. The new title & abstract is excellent. I have only two minor remarks:

* page 1/2: "...on one of the $42S_{1/2}$ sub level" -> levels

* The following paragraph is appearing twice ("Upon exposure to the optical field on resonance with the quadrupole transition...").

Reviewer #3 (Remarks to the Author):

The authors have carefully considered all comments and suggestions of reviewers. They revised the manuscript according to reviewer's recommendations (where it was possible), and clearly explained why some corrections or additional measurements were not done (like with the case of Hermite-Gaussian beams). For my opinion, the manuscript can be published now. I have two very small comments which concern more my "feeling of style".

1. Authors have added an explanation regarding the "beam's penumbra" (the text starting with "We call the beam's penumbra...") rather in the middle of manuscript. While the word "penumbra" was used already few times before. It might be confusing for the reader. I would move this explanation to the end of introductory part.

2. Regarding the title. (Again, I agree with Reviewer 1 that it is to a large extend the matter of taste). I like very much the idea of stressing the "Transfer of optical orbital angular momentum". But, for my opinion, the momentum is transferred not to an electron but to entire atom. Bound electron is a part of the atom. And we transfer the momentum to internal degrees (or states) of the atom.

Reply to reviewers.

We thank all reviewers for their final comments. Below we address the remaining questions.

Reviewer #1:

I would like to thank the authors for their comprehensive reply to the comments of all referees.

I am still slightly puzzled how the ion experiences a transition involving orbital angular momentum if does not have to be placed within a region that contains the singularity, but perhaps this issue will indeed become clearer in subsequent work as suggested by the authors.

Atoms can experience transitions involving more than just spin angular momentum when the rotation symmetry along the photons propagation axis is broken. This can happen, for example, if the atom is off axis with respect to the beam's center of if the magnetic field and propagation direction make a non null angle.

Apart from this point I think the authors have addresses all the comments from the referees and I can recommend publication in Nature Communications.

Reviewer #2 (Remarks to the Author):

In my opinion, the authors have answered all questions raised in the previous round satisfactorily. As I wrote before, I find this is a very nice piece of work. The new title & abstract is excellent. I have only two minor remarks:

* page 1/2: "...on one of the $42S_{1/2}$ sub level" -> levels

* The following paragraph is appearing twice ("Upon exposure to the optical field on resonance with the quadrupole transition...").

These mistakes have been corrected.

Reviewer #3 (Remarks to the Author):

The authors have carefully considered all comments and suggestions of reviewers. They revised the manuscript according to reviewer's recommendations (where it was possible), and clearly explained why some corrections or additional measurements were not done (like with the case of Hermite-Gaussian beams). For my opinion, the manuscript can be published now. I have two very small comments which concern more my "feeling of style".

1. Authors have added an explanation regarding the "beam's penumbra" (the text starting with "We call the beam's penumbra...") rather in the middle of manuscript. While the word "penumbra" was used already few times before. It might be confusing for the reader. I would move this explanation to the end of introductory part.

We considered this option but to do so we had to move too much information of the physics and results into the introduction. We believe, that as it is on reads through the paper building an idea of what the penumbra is, and finally when all the tools are there, a more precise definition is given.

2. Regarding the title. (Again, I agree with Reviewer 1 that it is to a large extend the matter of taste). I like very much the idea of stressing the "Transfer of optical orbital angular momentum". But, for my opinion, the momentum is transferred not to an electron but to entire atom. Bound electron is a part of the atom. And we transfer the momentum to internal degrees (or states) of the

atom.

Due to the alkaline-like nature of Ca^+ , the transferred OAM is indeed carried by the valence electron. Though formally the AM is transferred to the relative coordinate of the valence electron and the rest of the atom, the mass huge mass difference (more than four orders of magnitude) makes this association licit. Moreover, we did not entitle the work with "transfer of OAM to an atom" because we want to make clear the the OAM is NOT transferred to external d.o.f. Thus, the title is implying the correct physics and says in very short what we're doing.